# Novel Artificial Intelligence-Based Approaches for Ab Initio Structure Determination and Atomic Model Building for Cryo-Electron Microscopy

**DOI:** 10.3390/mi14091674

**Published:** 2023-08-27

**Authors:** Megan C. DiIorio, Arkadiusz W. Kulczyk

**Affiliations:** 1Institute for Quantitative Biomedicine, Rutgers University, 174 Frelinghuysen Road, Piscataway, NJ 08854, USA; 2Department of Biochemistry & Microbiology, Rutgers University, 76 Lipman Drive, New Brunswick, NJ 08901, USA

**Keywords:** cryo-electron microscopy, cryo-EM, deep learning, machine learning, artificial intelligence, AI, neural networks, AlphaFold2

## Abstract

Single particle cryo-electron microscopy (cryo-EM) has emerged as the prevailing method for near-atomic structure determination, shedding light on the important molecular mechanisms of biological macromolecules. However, the inherent dynamics and structural variability of biological complexes coupled with the large number of experimental images generated by a cryo-EM experiment make data processing nontrivial. In particular, ab initio reconstruction and atomic model building remain major bottlenecks that demand substantial computational resources and manual intervention. Approaches utilizing recent innovations in artificial intelligence (AI) technology, particularly deep learning, have the potential to overcome the limitations that cannot be adequately addressed by traditional image processing approaches. Here, we review newly proposed AI-based methods for ab initio volume generation, heterogeneous 3D reconstruction, and atomic model building. We highlight the advancements made by the implementation of AI methods, as well as discuss remaining limitations and areas for future development.

## 1. Introduction

Cryo-EM and single particle analysis (SPA) have become the preferred method for structure determination of biological complexes at near-atomic resolution [1,2,3,4,5,6]. Such level of detail not only provides crucial insight into molecular mechanisms employed by biological macromolecules, but also may facilitate the design and development of new drugs and therapeutics. The widespread adoption of cryo-EM is primarily credited to recent technological advances, including the introduction of direct electron detectors and improvements in computer hardware and image-processing software [7] that have enabled routine structure determination at high-resolution (reviewed in [8]). In contrast to traditional structural biology techniques, such as X-ray crystallography and Nuclear Magnetic Resonance (NMR), cryo-EM requires very small amounts of purified protein in the range of 0.1 to 5 mg/mL [9]. Additionally, SPA has proven to be a powerful tool for determining the structures of large and dynamic biological complexes that exhibit a range of compositional and conformational heterogeneity [10,11,12].

The goal of SPA is to calculate a high-resolution 3D structure from the noisy, 2D projections of the specimen produced on the direct electron detector by the beam of high energy electrons. Multiple software packages have been developed to facilitate this process [13,14,15,16,17,18,19,20], and a typical SPA workflow is described in Figure 1. However, several factors complicate image processing: (1) due to advancements in microscope optics, detector technology, and data storage hardware, a single cryo-EM experiment can generate thousands of micrographs with millions of particle images [21,22]; (2) biological assemblies are often dynamic and flexible molecular machines that adopt a variety of structural states; (3) cryo-EM micrographs have a low signal-to-noise ratio (SNR), and they are susceptible to artifacts, such as ice contaminations, radiation damage, and preferred orientation of imaged particles, which can obscure underlying structural information. While the application of traditional machine learning (ML) techniques has already enabled the automation of many stages of the workflow [23,24,25,26,27,28], ab initio reconstruction and atomic model building remain major hurdles in need of robust methodologies to minimize the extensive computational resources and manual user input they currently demand.

Recently, deep learning (DL) techniques have emerged as promising tools to tackle these aforementioned challenges because of their capability to learn complicated patterns and extract meaningful information from large and complex datasets. Although still in their infancy, various fully automated DL-based approaches have already proven useful for several image processing tasks, including particle picking [29,30,31,32,33,34,35], 3D reconstruction [36,37,38,39,40,41,42,43,44,45,46,47], local resolution estimation [48,49], and model building [50,51,52,53,54,55,56,57,58,59,60,61,62,63,64,65,66,67]. In this article, we explore the applications of new AI-based algorithms for two current bottlenecks of the cryo-EM image processing pipeline: ab initio reconstruction and de novo atomic model building. First, we briefly introduce the general architecture of several DL networks, followed by a discussion of conventional and DL-based techniques for ab initio 3D reconstruction. We then outline the existing challenges associated with atomic model building and present novel strategies for protein structure determination from cryo-EM maps. Finally, we discuss the utilization of DL-based protein structure prediction tools, including AlphaFold2 [68].

## 2. DL Algorithms

ML refers to a broad range of algorithms that learn patterns from data to make predictions, typically by using statistical and optimization methodologies. DL, on the other hand, is a subset of ML that utilizes so-called layered neural networks (NNs) to automatically learn, extract, and represent hierarchal patterns from complex and high-dimensional data, allowing for more sophisticated modeling and feature extraction (Table 1). DL algorithms have recently gained popularity for cryo-EM data processing due to the following factors. First, advances in data acquisition technology have enabled the collection of cryo-EM datasets containing tens of thousands of micrographs and millions of particle images. Such large datasets are required for training DL algorithms. Consequently, this has led to improved accuracy and performance of DL techniques for cryo-EM image processing. In addition, developments in computer hardware, particularly of graphic processing units (GPUs), enable computationally intensive DL algorithms to run faster and more efficiently. Moreover, significant innovations have been made in the design of DL architecture itself, notably by the development of convolutional neural networks (CNNs) that are ideally suited for image processing tasks.

DL algorithms consist of artificial neural networks (ANNs), powerful computational methods used to approximate non-linear functions. This ability enables DL algorithms to perform the various image processing tasks, including classification, pattern recognition, and optimization needed to recognize and model the complex relationships present within cryo-EM datasets. The general architecture of ANNs resembles the connected network of neurons in the brain [69]. ANNs are comprised of artificial neurons that utilize a mathematical function to convert input to output. To build such a network, the artificial neurons, also called nodes (Figure 2), are arranged in connected layers, where the output of one layer is the input to another layer. Each node in the system receives a collection of weighted inputs, and if the summed input surpasses a specific threshold determined by an activation function, the node transmits an output. Depending on how layers of the network are arranged, the ANN can learn high-level representations for classification tasks [70]. ANNs are trained by a process called backpropagation, in which the output of the ANN is compared to the expected outcome at each layer. In this iterative process, the calculated error is distributed from the network’s end to its beginning. As a result, the contributions of individual nodes to the error are determined based on their respective weights. Weighting parameters are then adjusted accordingly to minimize error, and after many iterations of training, predictions made by the ANN can improve.

CNNs are specialized ANNs that have gained significant popularity as unsupervised DL algorithms for 3D reconstruction [36,37,38,39,40,41,42,43,44,45] and atomic model building [50,51,52,53,54,55,56,57]; both are discussed in detail in later sections. CNNs utilize the convolutional operation to capture and analyze features from spatially organized data, such as 2D images and 3D volumes. The extracted information is then used to predict underlying properties of the data, for example, matching and classifying voxels of a 3D map as particular amino acid types [71,72]. The CNN architecture is made up of three connected layers: convolutional layers, pooling layers, and fully connected layers (Figure 3A). Convolutional layers serve to extract local features by convolving filters, also known as kernels, with the input data. If the input to the CNN is an image, the output of ‘sliding’ kernels over all pixels of the image is an array of image features, called a feature map. The feature map is then downsampled by the pooling layer to reduce both the spatial dimensions of the map and computational costs. By applying convolutional and pooling operations, the network learns to detect features at different scales and levels, capturing both low-level details and high-level semantic information. Lastly, the downsampled map is supplied to the fully connected layers, where learned features are combined and high-level predictions occur. By adding or “stacking” multiple convolutional, pooling, and fully connected layers on top of each other, CNNs can learn more nuanced information from the input data; as each layer in the network processes input and provides output to the next layer, the subsequent layers can build upon the representations learned in previous layers and extract higher-level features.

Another widely used DL method for cryo-EM data processing is the autoencoder (Figure 3B), a type of ANN that has been implemented in many recently developed ab initio [36,37,38,39,40,41,42,43,44,45] and heterogeneous reconstruction [38,39,40,42] algorithms. The autoencoder aims to accurately reconstruct its own input by encoding the input data into a lower-dimensional representation and then by decoding it back to its original form (Figure 3B). In the context of 3D reconstruction, the encoder network uses the input particle images to produce a lower-dimensional encoding, or latent variable, that serves as a compressed representation of the 3D structure. Here, the latent variable refers to a vector in which each element represents specific features of the underlying structure, such as particle poses. Based on the latent representation, the decoder generates reconstructed 2D images. The autoencoder is trained by minimizing the difference between the experimental and encoder-generated data. The Generative Adversarial Network (GAN) is yet another popular DL technique for 3D reconstruction [44,45]. GANs are unsupervised DL algorithms that use two CNNs, a generator and a discriminator, to model the distribution of the input data (Figure 3C). In the case of 3D reconstruction, the generator predicts a 3D map and produces 2D projections of the predicted volume. The discriminator then attempts to distinguish between the predicted 2D projections and the experimental projections (i.e., input particle images). GANs are trained with the aim of finding an equilibrium between the generator and discriminator, where the generator-produced images cannot be distinguished from the experimental images.

## 3. Conventional Approaches to Ab Initio Modeling and 3D Reconstruction

During 3D reconstruction, 2D particle images corresponding to different specimen views are computationally combined to obtain a 3D structure of the target macromolecule. Determining the true structure of the target specimen requires the accurate assignment of the experimental 2D images to particular 2D projections of the calculated 3D volume. Particle assignment, however, requires the calculation of the particle pose parameters (three Eulerian angles and two in-plane translations) that are not provided by the experimental images. This process is further complicated by several inherent factors of cryo-EM data: (1) artifacts are often present in samples and can be introduced during multiple stages of the EM workflow; (2) particle images have a very low SNR; (3) samples often have molecules of varying compositional and conformational states, and thus datasets contain 2D projections that correspond to multiple 3D structures (reviewed in [1]). Many existing 3D reconstruction algorithms use a low-resolution, initial map representing the best estimate of the target protein to assign projection directions to the experimental projections [73]. The projections and their corresponding poses are then used to update the reference map, and the process is iterated until the volume no longer improves. However, convergence to the correct solution is only guaranteed if the provided initial map is sufficiently close to that of the true structure. Furthermore, manual intervention is still required to evaluate the quality of the initial volume. Thus, obtaining a reliable initial model from any given dataset represents a challenge in cryo-EM.

There are several existing methods to obtain initial volumes for SPA, including the use of similar, previously determined structures [74], geometry-based techniques [75,76,77,78], and computational ab initio approaches [13,79,80,81,82]. In cases where a related structure is known, for example, a partial complex or homolog, it can be used as an initial model to estimate particle orientations by projection-matching [83] or by maximum-likelihood (MLM) approaches [13,14,84,85]. The goal of projection-matching is to determine poses for each particle image by measuring the cross-correlation coefficient between the experimental images and low-pass filtered, 2D projections of the input model. After orientations that yield the highest cross-correlation coefficient are assigned to each particle, a new 3D reconstruction is calculated that serves as the model for the next iteration of refinement. However, projection-matching algorithms are local optimizers and, thus, suffer from model bias, i.e., these approaches would arrive at an incorrect solution if the initial model were not close enough to the target structure. Alternatively, methods based on MLM for 3D reconstruction have been adopted by many different software packages [13,14,84,85]. Such techniques utilize an expectation-maximization algorithm to integrate over all possible probabilities of poses to calculate one or more 3D volume(s) [86]. While MLM methods have been successfully applied to extract multiple, heterogeneous structures from the same dataset [87,88], their convergence still requires one or more initial references that accurately reflect the underlying structural information.

Nonetheless, in many cases, prior structural information is unavailable. Thus, several approaches based on stage tilting have been developed to directly obtain an initial map from the sample. In cryo-electron tomography, particle orientations are directly measured by collecting multiple images of the same area over a range of microscope stage tilts [89]. The applicability and achievable resolution of this technique is limited by the radiation sensitivity of the sample subjected to multiple exposures, as well as the ability to collect high quality images at high tilt angles (e.g., 60 degrees). Similar specimen-tilting approaches, employed in the past for validation of structures obtained by SPA, include random conical tilt (RCT) [77,78] and orthogonal tilt reconstruction (OTR) [75,76]. In RCT, two images of the same area are collected, one with high sample tilt (e.g., 50 degrees) and the other untilted. Collecting titled pairs provides two of the three Euler angles that define particle pose for each particle image. The remaining angle representing in-plane translation is then computationally determined during the alignment of the untilted particles. However, the angular coverage of the specimen is limited by the inability to collect data beyond the maximum tilt angle (approximately 60 degrees), resulting in a “missing cone” of data in reciprocal space. This missing information directly corresponds to the particle orientations that were not imaged and introduces reconstruction artifacts, such as the elongation of the volume orthogonal to the tilt axis [90]. Additionally, the need to alter microscope stage tilt during image collection is cumbersome and poses a challenge to automation.

Methods have also been developed to computationally determine a reliable initial density map directly from the collected dataset. Several algorithms [79,82,91,92,93] apply the common line theorem to 3D reconstruction, which states that each 2D projection of the same 3D object share a common line in the 3D Fourier transform of the object. Therefore, common lines can be used to determine relative particle orientations between pairs of projections. Typically, this approach requires averaging of identical 2D views of the structure to increase the SNR. Thus, it can yield unreliable reconstructions in cases of low SNR. Other computational techniques include least-squares approaches [94], statistical weighting [79], and stochastic gradient descent (SGD) [13,95]. SGD relies on random initialization to generate an initial map using only particle images. However, because SGD algorithms are local optimizers, they may, in principle, yield incorrect initial maps [96]. An implementation of SGD in the program cryoSPARC [13] has become increasingly popular for calculating initial maps in the absence of prior knowledge of the target structure. This approach includes the capability of reconstructing multiple 3D maps to address sample heterogeneity. Nonetheless, all techniques described in this section are biased by the requirement for manual intervention to assess the quality of the calculated initial map(s) [97]. The recently developed method Xmipp Swarm consensus [98], available in Scipion [15], avoids user intervention by automatically calculating a consensus volume given a set of particles and initial maps obtained from different methods. This technique utilizes swarm optimization, a variation of SGD that incorporates momentum to accelerate convergence, to evolve the population of volumes towards a more globally correct solution [99].

It is important to note that when using any given method to calculate an initial map for subsequent 3D refinement, C1 symmetry should be used, unless other symmetry is already known. While applying symmetry constraints may increase the resolution of the final reconstruction, this reflects the increased number of particle images, rather than increased quality of the map. Enforcing symmetry in cases of no prior knowledge can produce artifacts or yield a wrong structure altogether. For example, applying the wrong symmetry to a dataset of the Magnesium Channel CorA resulted in maps with two additional incorrect densities [100].

## 4. DL Approaches to Ab Initio Volume Generation and 3D Reconstruction

Recent breakthroughs in microscope technology and advances in computer hardware and software have enabled the collection of increasingly large and complex cryo-EM datasets including hundreds of thousands of movie frames [21,22]. However, traditional methods for 3D reconstruction remain computationally expensive, time-consuming, and often fail to adequately address instances of structural variability. Thus, many methods have employed DL-based algorithms to determine particle poses, calculate initial 3D volumes, and elucidate different conformational states within the same dataset (Table 2). One of such widely used DL algorithms is the autoencoder, which has been implemented for both homogeneous and heterogeneous reconstruction by many programs, including cryoPoseNet [36], cryoAI [37], cryoDRGN [38,39,40], cryoFIRE [42], cryoVAEGAN [41], Atom-VAE [43], and 3DFlex [46]. As noted in the previous section, accurate 3D reconstruction requires the prior knowledge or calculation of particle poses. Currently, poses can be estimated by per-image pose searches or amortized inference methods. The former approach, employed by many traditional ab initio reconstruction and refinement algorithms [13,14,79,82,83,91,92,93], is not a DL-based strategy. In this approach, the posterior distribution of poses is computed for each experimental image independently, which is often computationally expensive. Amortized inference methods allow for faster pose estimation by amortizing or sharing computation across multiple instances through the use of a learned model. Amortized inference approaches leverage DL techniques to learn a model or parameters that capture the relationship between the input image and the corresponding particle pose. Following model training, these methods can predict the optimal pose for new images, significantly reducing the computational time required. Initially proposed by Ullrich et al. [101], many autoencoders apply an amortized inference approach for pose estimation. The program CryoPoseNet [36] was the first to use an autoencoder for ab initio 3D reconstruction with amortization over poses. CryoPoseNet encodes the input experimental images into a latent variable representing pose estimation. The predicted poses then become an input to a physics-based decoder. Such decoders incorporate the underlying physical principles and constraints of the cryo-EM imaging process, such as CTF parameters (i.e., defocus and angle of astigmatism) and projection parameters (i.e., Euler angles and translation shifts), to calculate a 3D reconstruction and generate corresponding 2D projections. However, currently cryoPoseNet has only been successfully applied to synthetic datasets. Because the cryoPoseNet image formation model operates in Fourier space and the 3D volume is stored in real space, each decoding step is slowed by the need to perform 3D Fourier and reverse Fourier transforms [36], constituting another limitation of the program. Furthermore, as shown by Ullrich et al. [101], amortized inference methods for pose estimation are highly susceptible to becoming stuck in local minima when the underlying 3D structure contains symmetries. Such minima trapping may result in symmetry artifacts, i.e., the resulting reconstruction displays the wrong symmetry. For example, the reconstruction of a synthetic hand by the autoencoder approach PoseVAE showed an incorrect planar symmetry [39]. CryoAI [37] has recently expanded upon cryoPoseNet, using a symmetric loss function to prevent local minima trapping where the map displays wrong planar symmetries. CryoAI has demonstrated the ability to perform ab initio reconstruction on experimental datasets, for example, 80S ribosome [37]. Both cryoPoseNet and cryoAI are currently only capable of producing a single consensus reconstruction. However, because the output from both programs include estimated particle poses and the 3D volume [36,37], the data can be used for subsequent structural refinements and/or heterogeneous reconstruction using other programs, for instance cryoSPARC’s heterogeneous refinement [13].

In addition to autoencoder approaches, methods using GAN architecture have been proposed for 3D reconstruction that avoid pose estimation entirely [44,45]. The program CryoGAN [44] utilizes a modified GAN framework for homogenous reconstruction that does not require an initial map. The goal of CryoGAN is to calculate the 3D volume whose distribution of simulated projections most closely match the experimental images. To accomplish this task, the generator is replaced with a cryo-EM physics-based simulator that imposes a mathematical model of the cryo-EM imaging procedure to produce images that resemble the real particle projections [44]. The cryo-EM physics-based simulator, for example, adds realistic noise to the simulated images that is extracted from areas of micrographs where no particles are present. However, because CryoGAN does not estimate poses, additional refinements cannot be applied to the resultant map, as such refinements would require the knowledge of particle poses [44]. CryoGAN developers have recently introduced Multi-CryoGAN [45], which allows for reconstruction of multiple volumes from a single dataset.

Because biological macromolecules are flexible molecular machines that exhibit an array of compositional and conformational states to execute a specific function, even biochemically purified cryo-EM samples often contain multiple structural states. The presence of multiple structural isoforms in the sample poses a challenge for 3D reconstruction, as averaging regions with structural variability often leads to blurred densities and limits the achievable resolution. Thus, it is important to calculate multiple maps in order to appropriately address structural variability. The past decade has introduced several advanced techniques aimed at elucidating a continuum of conformational states from cryo-EM data. One strategy is to approximate conformations as linear transformations of a known reference structure. Such dimensionality-reduction methods include principal component analysis (PCA)-based techniques [47,102,103,104,105] and normal mode analysis (NMA) approaches [106,107]. However, because these methods model variability as linear combinations of eigen volumes, they have limited ability to model complex, nonlinear motions. Other approaches aim to directly estimate a deformation field that describes how the initial reference structure must be modified to match the optimal structure for each individual image in the dataset [46,108,109]. For example, Herreros et al. utilized Zernike 3D polynomials to define a deformation field for approximating particle 3D conformations [108,109]. In the ‘hyper-molecules’ method [110], deformable molecules are represented as high-dimensional objects with additional degrees of freedom representing conformational space. Nonetheless, such techniques remain in the initial stages of development. For a more in-depth description of dimensionality reduction techniques for 3D reconstruction, see Singer et al. [111].

Another class of methods utilizes manifold embedding to uncover conformational heterogeneity [38,39,40,42,112,113,114]. In brief, these techniques encode images into a low-dimensional latent space, or manifold, to describe the conformational changes of the system. CryoDRGN [38] is one of the most widely used programs implementing manifold methods for heterogeneous reconstruction. By utilizing a modified autoencoder, cryoDRGN applies amortized inference to learn a distribution of conformational states within a dataset. However, the program assumes that poses are known and, thus, it requires a previously determined consensus reconstruction for initialization. Other versions of cryoDRGN, such as cryoDRGN-BNB [40] and cryoDRGN2 [39], implement ab initio reconstruction algorithms able to simultaneously estimate poses and different structural states. CryoDRGN2 has demonstrated the ability to perform homogeneous and heterogeneous reconstructions on both synthetic and experimental datasets, including the spliceosome and RAG1-RAG2 complex [39]. However, it does not apply an amortized inference approach for pose estimation. Rather, this method utilizes an exhaustive pose search in the 5-D space of rotations and in-plane translations for each individual image [39]. Such technique, in turn, often poses a computational bottleneck.

To efficiently sample the conformational space of the biological specimen, cryo-EM datasets have to grow increasingly large, making apparent the need for efficient reconstruction techniques whose runtimes scale with dataset size. Another program from the developers of cryoDRGN, cryoFIRE [42], is the first amortized inference approach for ab initio, heterogeneous reconstruction applicable to large, experimental datasets. CryoFIRE performs joint amortization of poses and conformational states, and it utilizes a physics-based decoder to input images to a conformational manifold [42]. This autoencoder architecture allows for significantly faster image processing than existing methods, for example, heterogeneous ab initio reconstruction using five million particle images can be completed in two hours [42]. However, this approach demonstrates lower accuracy in translation estimation compared to the per-image pose searches employed by cryoDRGN-BNB [40] and cryoDRGN2 [39], which can result in lower accuracies of particle alignments, subsequently leading to lower resolution reconstructions [42]. Furthermore, interpreting the conformational space provided by manifold methods remains challenging, as it might not reflect the conformational space in vivo.

While the techniques described above have demonstrated progress in elucidating multiple conformational states present within a cryo-EM dataset, they do not provide fully quantitative information about the dynamics of the target system [115,116]. Furthermore, map densities that tend to correlate with dynamics information (i.e., flexible regions) often exhibit lower local resolution. Thus, algorithms have been developed that incorporate cryo-EM maps into molecular dynamics (MD) simulations to provide dynamics information consistent with both the experimental data and the physical–chemical constraints of the biomolecule (e.g., bond length, dihedral angles, etc.) [116,117,118]. The program DEFMap [118] is a DL-based strategy designed to predict dynamics associated with atomic fluctuations within cryo-EM maps. DEFMap first performs all-atom MD simulations using PDB structures derived from EM maps to calculate the root mean squared fluctuation (RMSF) representing atomic fluctuations. The program then employs a 3D CNN to learn the relationship between the cryo-EM map and the MD-derived RMSF values, capturing the 3D patterns of the experimental data that represent protein dynamics [118]. Another program, CryoFold [119,120], integrates cryo-EM density data with MD simulations and other modeling tools to generate the most probable ensemble of atomic structures. CryoFold has successfully produced models of the CorA channel and ATP synthase [119]. However, the program requires density maps with 5 Å resolution or better [119]. Furthermore, because cryo-EM maps are typically calculated using only a filtered subset of the experimental images, the techniques described above often do not sample the complete conformational landscape of the target system [117].

Another strategy to characterize conformational states and their probability distribution is to directly infer atomic models from the experimental images, bypassing the 3D reconstruction process entirely. Rosenbaum et al. utilized a VAE to infer a continuous distribution of atomic models and poses directly from particle images [43]. Other approaches combine Bayesian inference and physical structure-sampling tools to generate model ensembles that match the experimental cryo-EM data, including BioEM [121,122], cryoBIFE [123], and ensemble reweighting [124]. cryoBIFE uses a path collective variable to generate free-energy profiles for molecules directly from particle images along with its uncertainty [123]. Expanding upon cryo-BIFE, the ensemble reweighting approach developed by Tang et al. extracts ensemble densities directly in atomic coordinate space using cryo-EM particle images [124]. This approach first generates an initial guess of the system’s conformational probability landscape using prior ensembles from protein structure prediction tools or MD simulations. The landscape is then reweighted by comparing the experimental cryo-EM images with the conformations sampled from the initial distribution. The reweighted ensemble can then be used to produce ensemble averages or calculate free-energy landscapes [117]. While this approach can potentially be applied to flexible systems for which a reliable 3D reconstruction cannot be obtained, it has yet to be applied to experimental datasets. For a more in-depth discussion of methods for inferring the probability distribution of conformations from cryo-EM data, see Tang et al. [117].

Taken together, significant progress has been made in facilitating 3D reconstruction from cryo-EM data with the application of DL-based algorithms. Several methods have been applied to homogenous reconstruction [36,37,44] that eliminate the requirement for an initial model. However, so far, these methods have only been applied to synthetic [36,44] and high-quality, publicly available experimental datasets [37]. The successful application of DL-based methods has yet to be demonstrated for structure determination from novel cryo-EM datasets. Further developments are also needed to match and surpass the resolutions achievable by conventional reconstruction methods [44]. As datasets grow in size to accommodate more complex systems, DL-based ab initio reconstruction methods show great promise in elucidating distributions of heterogeneous structures [38,39,42,45]. These techniques overcome the computational bottlenecks that plague traditional approaches by utilizing amortized inference. However, developments are needed to increase the accuracy of pose estimation [42]. Very recently, several techniques have been proposed to combine cryo-EM data with physical structure-sampling tools, such as MD simulations, to provide dynamics information [116,118,119,120] and characterize conformational landscapes of the target systems [121,122,123,124].

**Table 2 micromachines-14-01674-t002:** Summary of recently developed AI-based methods for ab initio 3D reconstruction. Many techniques utilize a variational auto-encoder (VAE) architecture. Unlike conventional autoencoders that compress input images into a fixed encoding, VAEs learn a probability distribution of encodings per image, enabling the generation of new data through sampling.

Program	AI architecture	Advantages	Limitations
cryoPoseNet [36]	Auto-encoder	Amortized inference over poses Reconstruction can be further refined by other programs	Reconstructs single, consensus volume Volume susceptible to trapping in local minima Only demonstrated on synthetic datasets
cryoAI [37]	VAE	Amortized inference over poses Reconstruction can be further refined by other programs Implements symmetric loss function to avoid local minima trapping in pose estimation Demonstrated on experimental datasets	Reconstructs single, consensus volume
cryDRGN-BNB [40]	VAE	Amortized inference over conformations Reconstructs manifold for continous and discrete conformation distributions	Per-image pose search is computationally expensive Scales poorly with larger datasets Fails to reconstruct high-quality volumes from experimental datasets Difficult to interpret conformational landscape
cryDRGN2 [39]	VAE	Amortized inference over conformations Reconstructs manifold for continous and discrete conformation distributions Demonstrated on experimental datasets More accurate pose estimation than later version [42]	Per-image pose search is computationally expensive Difficult to interpret conformational landscape
cryoFIRE [42]	VAE	Joint amortized inference over conformations and poses Reconstructs manifold for continous and discrete conformation distributions Reconstruction significantly faster than previous versions [39,40]	Pose estimation can be inaccurate, limiting resolution of reconstructions Difficult to interpret conformational landscape
spatial-VAE [125]	VAE	Estimates translations and in-plane rotations	Does not directly perform 3D reconstruction
Multi-CryoGAN [45]	GAN	Reconstructs manifold for continous and discrete conformation distributions Avoids pose and conformation estimation for each projection	Only demonstrated on synthetic datasets Reconstruction cannot be further refined Lower resolution than traditional methods Difficult to interpret conformational landscape
cryoVAEGAN [41]	VAE, GAN	Jointly estimates in-plane rotation and CTF parameters	Does not directly perform 3D reconstruction

## 5. Conventional Approaches to De Novo Atomic Model Building and Refinement

Recent developments in the field of cryo-EM have resulted in the improved quality and resolution of Coulomb maps deposited to the Electron Microscopy Data Bank (EMDB). For instance, in 2022, 2734 maps with 4 Å resolution or better were deposited to the EMDB, compared to just 90 maps in 2015 [126]. In high-resolution maps (≤3 Å), one can directly observe residue side chains and distinguish between different side chain rotamers [127]. De novo model building for high-resolution maps is relatively straightforward and can be accomplished using tools originally developed for X-ray crystallography and adopted for cryo-EM, such as Phenix [128] and Coot [129,130] (Figure 4). Nonetheless, it is important to note that this process still requires manual user intervention and is sometimes very labor-intensive. At intermediate resolution (4–6 Å), map topology becomes increasingly difficult to resolve, making atomic model building a nontrivial task. At such resolutions, one may be able to recognize the pitch of helices, but individual beta strands and residue sidechains may not be discernable. Furthermore, resolution is rarely uniform throughout a given map, limiting the building of an accurate atomic model from the entire map.

If available, a previously determined structure, for example, by X-ray crystallography or NMR, with a homologous sequence can be used as a template for model building [131]. In such cases, the existing model is docked in the EM map and refined to maximize the model agreement with both molecular geometries and the EM densities. Techniques such as rigid fitting [132,133] and flexible fitting [134,135,136] can be used to fit the template structure in the EM map. These approaches use a scoring function to measure the fit of the atomic model to the EM map. However, if no homologs are available, de novo modeling must be performed to produce an atomic model based on the discernable map features and protein sequence. This process involves determining the densities that correspond to the protein backbone and assigning sequence to build the model. Traditional de novo modeling methodologies utilize physics-based optimization algorithms to build atomic models from intermediate resolution maps [72,128,137,138,139]. These programs, including EM-Fold [137], Rosetta [138], Gorgon [139], and MAINMAST [72], iteratively optimize an atomic model by minimizing an energy function that includes terms describing physical forces, such as steric clashes, bond lengths, and electrostatic interactions. However, such algorithms have high computational costs and often heavily rely upon manual intervention for accurate model building.

After calculating an initial model, refinement methods guided by experimental data play a crucial role in improving the model quality. Such methods aim to optimize the fit of the structural model to the experimental map while maintaining geometry restraints [127]. The program MDFF [140,141] employs a MD-based approach and is one of the most popular refinement methods, with successful applications of model refinement of ion channels [142] and various structures from SARS-CoV-2 [143,144]. MDFF refines structures through MD simulations in the presence of external constraints derived from the experimental volume that are related to the density gradient of the EM map and its Coulomb potential. These constraints then guide the atomic model towards the position that best fits the experimental densities [141]. Another method, Flex-EM [145], optimizes atomic positions with respect to a scoring function that includes the cross-correlation between the map and model, as well as stereochemical and non-bonded interaction terms. In Flex-EM, the model is divided into rigid bodies that undergo heuristic optimization by a Monte Carlo search, conjugate-gradients minimization, and simulated annealing MD [145]. The program EM-refiner [146] employs replica-exchange Monte Carlo (REMC) simulations to refine atomic models. Here, refinement simulations incorporate data from the EM map with physics- and knowledge-based force fields to direct flexible fitting of the backbone structures.

There are several hurdles that limit the accuracy of derived models from intermediate-resolution EM maps, which are the most prevalent. As target structures often contain multiple, related subunits, many existing model building and refinement algorithms require segmentation of the map into sub-densities representing individual subunits or nucleic acid chains. However, segmentation becomes increasingly difficult if areas of the map vary in resolution, and virtually impossible in cases where the map quality is too poor to resolve subunit interfaces [127]. Furthermore, maps are often noisy due to several factors, including the low SNR in cryo-EM data, particle misalignment, or the presence of structural heterogeneity and preferred orientations. Signal contributed by noise may manifest as random fluctuations or distortions in the map that can interfere with the optimization of the measured fit between the model and the Coulomb map [62]. Additionally, there is currently no infrastructure to adjust models of flexible macromolecules that undergo continuous conformational changes [1]. The recent integration of AI-based approaches to the de novo model building toolbox has the potential to significantly alleviate these limitations.

## 6. AI-Based Approaches to De Novo Model Building

Early applications of AI methods to model building utilized conventional ML techniques such as k-nearest neighbor (k-NN) [147], k-means clustering [148], and support-vector machines (SVM) [149]. These approaches have been successful in identifying SSEs [147,149] and modeling simplified backbone structures [148]. For example, the program RENNSH [147] identifies α-helices in EM maps by representing each voxel as a spherical harmonic descriptor and using a nested k-NN framework to classify α-helix voxels. Another ML approach, SSELearner [149], first learns from maps deposited in the EMDB and then applies an SVM classifier to detect both α-helices and β-sheets in intermediate-resolution density maps. Pathwalking [148] is a de novo model-building approach that uses a combination of the traveling salesman problem and k-means clustering to build a Cα model. Nonetheless, these programs do not guarantee convergence to the correct solution and are not capable of building complete atomic models [71,150].

DL has recently driven remarkable advancements in structural biology, including the development of programs for automated de novo atomic model building from cryo-EM density maps (Table 3). This breakthrough is largely attributed to increased computational power, quality of EM images, and the number of available high-resolution cryo-EM structures for model training [150]. Recently, many CNN-based approaches have been applied not only for automatic assignment of SSEs [50,51,52,53] and backbone chains [53,54,55,56], but also to detect individual amino acids [56,57], thus generating complete de novo models [53,54,55,56,57] (Table 3). Below, we highlight some representative programs utilizing CNN architectures, followed by a description of methods that apply alternative DL frameworks to model building.

AAnchor [56] is the first program with 3D CNN architecture to identify individual amino acids in EM maps. This technique utilizes a classification CNN to locate and label residues, known as anchors, with highest confidence within a defined voxel size [56]. However, AAnchor is currently limited to maps with 3.1 Å resolution or higher and on average, the fraction of anchors detected ranges from 10 to 20% depending on map resolution [56]. Another program, A^2^-net [57], uses a CNN framework to identify residues and their poses followed by a Monte Carlo Tree Search (MCTS) strategy to link the 3D coordinate system of amino acids into a complete peptide chain. Compared to its automated counterparts, including Rosetta and others [151,152], A^2^-net converges to a solution significantly faster. An atomic model for a protein with one thousand amino acids can be derived in minutes [56], compared to several hundred hours in Rosetta [152]. The latest version of A^2^-net, CryoNet (https://cryonet.ai (accessed on 10 May 2023)), has been applied for model building of cryo-EM maps of the human minor spliceosome [153] and the *Acidobacteria* homodimeric reaction center bound with cytochrome c [154]. CR-I-TASSER [54] is a fully automated, hybrid method that combines a 3D-CNN to build Cα trace models with multithreading algorithms to identify homologous Protein Data Bank (PDB) templates for guided structure assembly. Nonetheless, the generated CR-I-TASSER model directly relies on the accuracy of the Cα trace prediction, which authors found to decrease with lower resolution maps [54]. Additionally, this technique requires prior segmentation of the experimental map, further limiting its applicability to low-resolution density maps. Although CNNs remain one of the most popular AI architectures for atomic model building and are the most direct DL method for learning features from density maps, they are not without limitations. While CNNs are considered translation-equivariant, meaning translating the input data will result in translated output, they lack rotational invariance, meaning they do not behave consistently for input data of varying orientation. Moreover, the localization of the convolution mechanism narrows the CNN’s receptive field [155,156], potentially limiting their ability to capture global features and dependencies within cryo-EM maps.

In addition to standard CNN framework, many de novo model building programs have adopted the U-Net architecture (Table 3). The U-Net is a type of CNN commonly used to segment or classify pixels in an input image [157]. In the U-Net architecture, an encoder first downsamples the input image and extracts features, and then a decoder network restores the image back to its original dimensions. Through the downsampling and upsampling operations, U-Nets may be more effective at extracting both low-level and high-level features than standard CNNs. Moreover, U-Nets incorporate so-called skip connections that, as the name implies, may skip any given layer in the neural network to provide direct connections between different layers of the network. In the U-Net framework, skip connections enable the direct propagation of feature maps from the encoder to the decoder. This unique architecture preserves fine-grained details and facilitates better information flow throughout the network. Various programs have employed a 3D U-Net framework for SSE identification [58,59] and complete de novo model building [60,62]. DeepTracer [60] is one of the most popular, fully automated model building programs that utilizes a 3D U-Net architecture to predict locations of SSE elements, backbone atoms, and individual amino acids within EM maps. Evolved from Cascaded-CNN [53], DeepTracer employs a series of four U-Nets to perform distinct tasks: locate amino acid positions, locate the protein backbone, identify SSEs, and identify individual residues [60]. Pfab et al. applied DeepTracer to a set of coronavirus-related cryo-EM maps (Figure 5), and found that, on average, 84% residues match with the corresponding deposited PDB structures [60]. Furthermore, DeepTracer has been utilized to build a variety of atomic models from EM maps, including the human small subunit processome [158], Chikungunya virus replication complex [159], and human caveolin-1 complex [160]. While DeepTracer demonstrates more accurate Cα prediction than its counterparts [72,128,161], it is currently only applicable to maps with 5 Å resolution or higher. The latest version, DeepTracer-2.0 [61], also has the ability to model nucleic acids.

Other DL architectures for de novo model building include the Graph Neural Network (GNN), Recurrent Neural Network (RNN), Residual Neural Network (ResNet), and Long- and Short-Term Memory network (LSTM) (Table 3). For example, Structure Generator [55] is a fully automated pipeline that utilizes 3D-CNN, GCN, and LSTM algorithms to produce a protein structure based on amino acid identities and locations. Structure Generator [55] first uses a 3D-CNN, named RotamerNet, to locate amino acids and assign their rotameric orientations within the map. Using the Cα coordinates defined by RotamerNet, the GCN generates a contact map and connects Cα positions located within 4 Å. Lastly, the bidirectional LSTM labels the candidate amino acids and ensures the consistency of the assignment with a provided protein sequence to yield a structural model [55]. While Structure Generator has demonstrated high prediction accuracy using simulated datasets [55], it has not yet been applied to experimental maps. Additionally, the program is limited to modeling protein sequences with less than 700 amino acids [55].

DL techniques have provided much needed insight into the automation of atomic model building. Several approaches have demonstrated capabilities to construct full atomic models, but their applicability is generally limited to maps with 5 Å or higher resolution [60,61,64,163]. Furthermore, while several methodologies are able to precisely identify and locate individual atoms, it remains a challenge to construct a full peptide chain without violating geometric and stereochemical restraints [65,71]. Different solutions have been proposed to remedy this problem, including the program ModelAngelo [64,163] that utilizes sequence data and prior information of protein geometries to refine the protein chain geometry. Another software, CR-I-TASSER, attempts to overcome this limitation by integrating Molecular Dynamics simulations with DL model building [54]. New strategies have also been proposed to integrate DL frameworks with protein structure prediction techniques, such as AlphaFold2, discussed in detail in the following section.

**Table 3 micromachines-14-01674-t003:** Summary of recently developed AI methods for de novo atomic model building.

Program	AI Architecture	Advantages	Limitations
Emap2Sec [50]	CNN	Identifies SSEs Demonstrated on experimental maps High accuracy for intermediate resolution maps	Does not place α-helices and β-sheets in detected regions Voxel-based approach fails for large EM maps
Emap2Sec+ [52]	ResNet	Identifies SSEs and nucleic acids in maps 5–10 Å Improved accuracy in protein SSE detection compared to Emap2Sec [50]	Voxel-based approach fails for large EM maps
CNN-based [51]	CNN	Identifies SSEs	Does not place α-helices and β-sheets in detected regions No results for experimental maps
Haruspex [58]	U-Net	Identifies SSEs and nucleic acids in maps Applied to experimental and simulated maps	Only applicable to maps ≤4 Å False positives for helices, sheets or RNA/DNA Misclassifies semi-helical elements, β-hairpin turns, polyproline residues as α-helices
EMNUSS [59]	U-Net	Identifies SSEs Applied to experimental and simulated maps	Incorrect predictions on atypical density volumes; narrow receptive field
AAnchor [56]	CNN	Identifies amino acids	Only applicable to maps ≤3.1 Å Only detects 10–20% amino acids on average
A^2^-Net [57]	3D-CNN, MCTS	Identifies amino acids Model with 1000 amino acids can be derived in minutes Applied to experimental maps Fully automated	Cannot identify ligands
Structure Generator [55]	3D-CNN, GCN, LSTM	Identifies amino acids and rotamer orientation, builds full protein chain	Limited to protein sequences <700 amino acids No results for experimental maps
CR-I-TASSER [54]	3D-CNN, I-TASSER	3D-CNN predicts Cα atoms used for I-TASSER; generates full structure	Cα trace prediction accuracy dependent on resolution Requires prior map segmentation
DeepTracer [60,61]	3D U- Net	Locates amino acid positions and protein backbone Identifies SSEs and amino acids Automated Lastest version models nucleic acids	Only applicable to maps ≤5 Å Only builds atoms for main chains
DeepTracer ID [63]	3D U-Net, AlphaFold2	Uses DeepTracer to build model and searches against AlphaFold2 library for refinement Identifies individual proteins in density map; does not require protein sequences to be known a priori Does not require high accuracy from AlphaFold2 prediction	Limited to proteins >100 amino acids for succesful AlphaFold2 prediction matches Only applicable to maps ≤4.2 Å
EMBUILD [62]	U-Net, AlphaFold2	Constructs main chain map; fits AlphaFold2 predicted chains into the map	Only builds atoms for main chains
ModelAngelo [64,163]	CNN, GNN	Builds complete atomic model Better RMSD and sequence prediction results than DeepTracer	Only applicable to maps ≤3.5 Å Requires protein sequences are known
DeepMM [164]	CNN	Predicts Cα positons; identifies SSEs and amino acids Applied to experimental maps	Cannot model nucleic acids Residue matching accuracy dependent on map resolution
DEMO-EM [165]	ResNET, I-TASSER	Builds complete atomic model for multi-domain proteins Only requires protein sequence Fully automated	Requires prior map segmentation Individual domain models calculated without restraints from density data

## 7. Applications of AI-Based Protein Structure Prediction to Atomic Model Building

The field of protein structure prediction has been transformed by significant developments in DL algorithms, exemplified by AlphaFold2 [68], RoseTTaFold [166], ESMFold [167], and I-TASSER-MTD [168]. Given only protein sequences, such programs employ a combination of NNs and novel algorithms to predict atomic 3D structures, which in turn provide complementary data to experimental methods. Of particular note, AlphaFold2 [68], a publicly available program developed by Google DeepMind, has shown unprecedented levels of accuracy in predicting atomic models [68,169,170]. AlphaFold2 outperformed all other prediction methods in the 14th edition of the Critical Assessment of Structure Prediction (CASP), a blind test for structure prediction of experimentally solved structures that have not yet been made publicly available. AlphaFold2 models had a median backbone accuracy of 0.96 Å RMSD, whereas the next best performing method had a median backbone accuracy of 2.80 Å RMSD [68]. Using the amino acid sequence as input, AlphaFold2 first performs a Multiple Sequence Alignment (MSA) and tries to identify proteins with similar structures, known as templates. Based on the templates, AlphaFold2 generates a “pair representation” model, indicating which amino acids are likely to be in contact with each other [171]. The program then employs a transformer NN, called the Evoformer, to refine, exchange, and extract information from the MSA and pair representation. The extracted information is used by the structure module, a second NN, to construct an atomic model [68]. The AlphaFold Protein Structure Database [172], hosted at the European Bioinformatics Institute–European Molecular Biology Laboratory (EMBL-EBI), provides access to over 200 million AlphaFold2 models, covering the majority of sequence entries in UniProt. Furthermore, computed models using AlphaFold2 as well as other prediction methods are available through ModelArchive (modelarchive.org). As of 2022, over one million computed models were also accessible through the Research Collaboratory for Structural Bioinformatics Protein Data Bank (RCSB PDB) [173].

Predicted structures can serve as starting models for deriving atomic coordinates from cryo-EM maps [174,175,176,177,178]. In addition, several methods have been proposed to dock and fit predicted structures into density maps [62,66,67]. In cases of multi-domain and multi-component complexes, one approach is to first use AI-prediction tools to generate structures of sub-complexes and then assemble them into higher-order structures [168,179,180]. For example, the protein structure prediction and modeling tool I-TASSER-MTD [168] predicts structures of individual domains and then uses the EM density to guide the construction of a complete atomic model. Assembline [180] is another software package that offers an integrative pipeline for model building that satisfies the constraints of predicted structures and cryo-EM maps. Assembline has been used to assemble full atomic structures of multiple protein complexes [178,181,182], including the human nuclear core complex [178]. In this study, Mosalaganti et al. fit models of individual subunits and sub-complexes generated by AlphaFold2 [68] and ColabFold [183], respectively, into the density map, followed by structure assembly with Assembline [180]. Notably, the model of the N-terminal domain of NUP358 predicted with AlphaFold2 was in better agreement with the cryo-EM map than atomic coordinates from previously determined X-ray structures [178]. Furthermore, the predicted models not only showed the same subunit interactions previously reported by the crystal structures, but also revealed new interactions [178].

Another approach is to combine structure prediction tools with DL-based model building techniques to aid in the construction and refinement of atomic models [62,63,64,65,66,67,165]. The program EMBUILD [62] utilizes a U-Net to first generate a so-called main-chain probability map including potential solutions and probabilities reflecting the agreement of the main chain assignments with experimental densities. AlphaFold2 [68] predicted models of individual chains are then fit into the map, and a scoring function is used to measure how well the fitted chain matches the main-chain probability map. The final model is the combination of fitted chains that yield the highest scoring function among different combinations of fitted positions [62]. He et al. [62] applied EMBUILD to 47 intermediate-resolution maps and built models with an average template modeling score (TM-score) of 0.909 and RMSD of 2.85 Å, outperforming other state-of-the-art methods, including phenix.dock_in_map [184] and gmfit [185,186]. Another program, DEMO-EM [165], utilizes DL-based inter-domain distance maps to facilitate the fully automated assembly and refinement of individual domains into full-length structures. When applied to a benchmark set of multi-domain proteins with medium- to low-resolution maps, DEMO-EM generated models with correct inter-domain orientations in 97% of cases, outperforming MDFF and Rosetta [165]. Nonetheless, DEMO-EM, along with the other programs described in this section, utilize models generated for each subunit independently without taking into consideration the conformational changes that may occur upon subunit binding [64].

Other algorithms leverage structure prediction tools to identify proteins within cryo-EM maps. With advancements in image processing techniques, cryo-EM can now be applied directly to cellular extracts. However, determining the specific proteins within the resulting map often requires the application of other complementary techniques, such as tandem Mass Spectrometry. The program DeepTracer-ID [63] combines DeepTracer [60,61] and AlphaFold2 to directly identify the proteins present in a cryo-EM map. DeepTracer-ID first utilizes DeepTracer, as described above, to build an atomic structure, and then uses this model to search the AlphaFold2 library containing all predicted structures from any given organism. Identified structures are used to iteratively refine the atomic model. In a blind test of 13 experimental maps, DeepTracer-ID identified the correct proteins as top candidates [63]. DeepTracer-ID has also been successfully used for protein identification in cryo-EM maps of a white spot syndrome virus capsid [187] and archaeal surface filaments [188]. Nonetheless, this approach is only applicable for maps with 4.2 Å or higher resolution. In addition, accurate sequence-based alignment of the DeepTracer model and AlphaFold2 predicted structure requires proteins with a sequence longer than 100 amino acids [63]. Similarly, the program DeepProLigand [65] predicts protein ligand interactions by employing DeepTracer and using other known protein-ligand structures available in AlphaFold2 library or available through the RCSB PDB.

While integrative approaches described in this section represent a promising tool for atomic model building, AI structure prediction methodologies suffer from several limitations. Assessing the quality of predicted AlphaFold2 structures is a nontrivial task. Although quality metrics accompany each AlphaFold2 model, these scores are also predictions that may contain errors [189]. Thus, manual inspection is still required to evaluate the model in the context of the experimental map, as even high-confidence predictions can be modeled incorrectly [190]. Furthermore, as AlphaFold2 relies on experimentally determined structures deposited in the PDB, the algorithm may produce incorrect folds when there is a lack of homological structures [70]. Further developments are needed to improve the prediction accuracy of mutated residues, regions involved in ligand binding, and dynamic interactions [189,190].

## 8. Conclusions

Over the past decade, technological advances have transformed single-particle cryo-EM from low-resolution “blobology” [7] to the preferred technique for structure determination at near-atomic resolution. The recent adoption of DL techniques has made significant contributions to the cryo-EM image processing pipeline, particularly in the areas of 3D reconstruction and atomic model building. As the size of cryo-EM datasets become larger to efficiently sample more complex and dynamic specimen, there is an increasing need for efficient ab initio reconstruction methods. While a variety of DL-based techniques have been applied to generate initial 3D density maps, including auto-encoders and GANs, these approaches are still in the early stages of development, and many have only demonstrated success on synthetic datasets [36,45]. Further improvements are needed for DL-based algorithms that leverage amortized inference to accurately tackle heterogeneous ab initio reconstruction without compromising computational resources. Additionally, the interpretation of the conformational landscapes generated by DL-based reconstruction algorithms remains challenging, as these techniques lack benchmarks and standardized stand-alone validation measures. The current methods rely on the user to manually inspect the manifold and reconstruct a subset of volumes at individual points. Cases of discrete heterogeneity are often unambiguous, as distinct structures frequently manifest as clusters of points in the manifold that allow users to generally infer the number of states and their populations [38]. However, due to the nonlinear relationship between the latent space and the distribution of volumes, distances between points in the landscape have no straightforward, real-world meaning. While these models aim to organize their latent space such that structurally related conformations reside in close proximity, there is no physically interpretable metric that relates the trajectory between different conformations in the manifold [42]. Techniques like PCA can help visualize the directions of significant variability; however, assigning biological significance to this motion is ill-advised. The validity of all inferred conformational states requires a combination of experimentally derived structures coupled with other biophysical and biochemical tools that probe dynamics of biological systems. Importantly, DL-based techniques have also proven useful for constructing atomic models from intermediate-resolution cryo-EM density maps. A number of methodologies have been developed to extract features from 3D volumes, for instance SSEs, Cα backbones, and individual amino acids. The most successful of these approaches employ CNN and U-Net frameworks. As the field of DL continues to rapidly develop, the application of more complex DL frameworks, such as transformer models [191], have yet to be explored for de novo atomic model building [71,150]. In addition, several tools that utilize structures from AlphaFold2 and other protein prediction software to guide structure assembly have been recently proposed. However, there remains a need for more end-to-end, fully automated techniques that utilize complementary data, such as amino acid sequences and predicted protein structures, for more accurate structure modeling.

## Figures and Tables

**Figure 1 micromachines-14-01674-f001:**
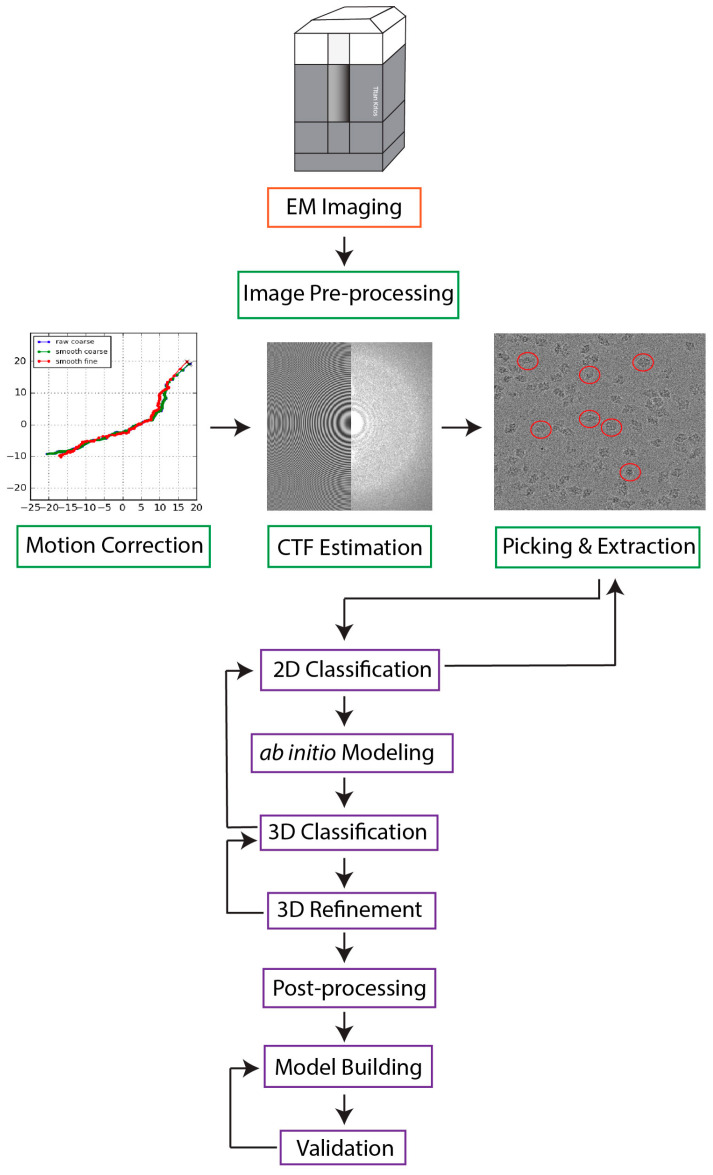
A typical SPA workflow. Movies collected with an electron microscope are first motion-corrected. In this step, frames are aligned and averaged to account for beam-induced motion, which increases the SNR of images. The resultant micrographs undergo contrast transfer function (CTF) estimation to calculate the effects of defocus and microscope aberrations. This step is followed by particle picking and extraction, in which particles are selected and extracted from micrographs. The extracted particles are sorted based on orientation into discrete, 2D classes, and the user removes classes containing non-particles, noise, and artifacts. Such filtered particle stacks are used to generate one or more low-resolution ab initio reconstructions that are iteratively refined through 3D classification and 3D refinement to yield final Coulomb potential maps. Given sufficient map resolution and quality, atomic models can be built and validated.

**Figure 2 micromachines-14-01674-f002:**
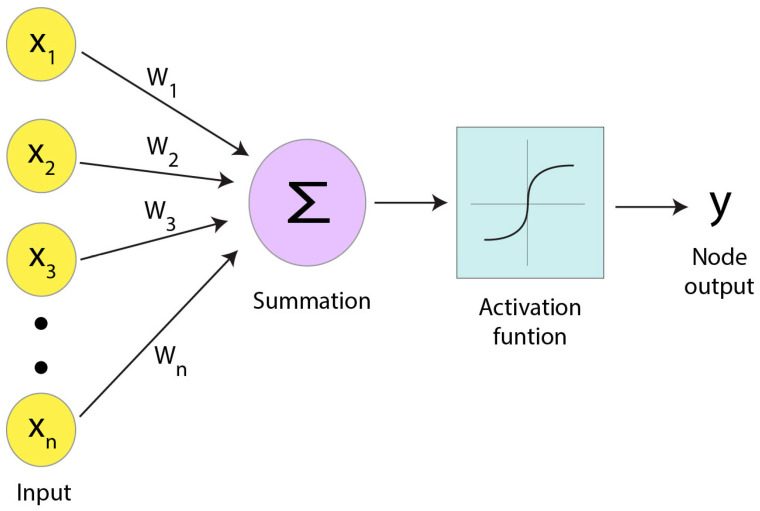
Schematic of a basic ANN node. The node receives inputs (x_1_, x_2_, x_3_, …x_n_) from a preceding layer in the network (yellow). Each input is multiplied by a corresponding weight (w_1_, w_2_, w_3_, …w_n_, respectively). The node computes the sum of the weighted inputs and applies an activation function that generates the cumulative output of the node (y). This output is then transmitted to subsequent network layers for further processing.

**Figure 3 micromachines-14-01674-f003:**
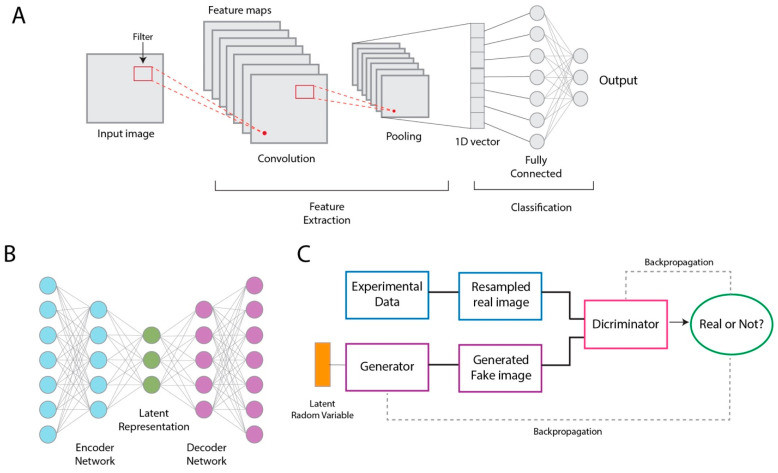
Schematic of different DL architectures. (**A**) Convolutional Neural Network. In a CNN, a convolutional filter, or kernel, (red box) slides over the input data and extracts different features of the data, generating a corresponding feature map. Many maps are generated. These feature maps are then downsampled in the pooling layer to produce a “flattened” 1D image vector, which subsequently serves as the input to the fully connected layer where classification occurs. (**B**) Autoencoder. In the autoencoder architecture, the encoder NN (blue circles representing individual nodes in the network) transforms input data to a lower-dimensional, simplified latent representation (green circles). The decoder network (shown in purple) converts the latent representation back to the original dimension and form of the input. (**C**) Generative Adversarial Network. In a GAN, the generator aims to produce synthetic images that closely resemble the input data. The generator initiates this process with a latent variable (orange), which consists of a vector of random values. By adjusting the values of the latent variable, the generator can produce an array of synthetic outputs, thereby exploring different variations in the generated samples. The real, experimental images and the generated images are both provided as inputs to the discriminator network, which evaluates whether the images are real or not. During training, both networks update their weights based on the generator’s ability to produce realistic images and the discriminator’s ability to accurately decipher between the real and synthetic images.

**Figure 4 micromachines-14-01674-f004:**
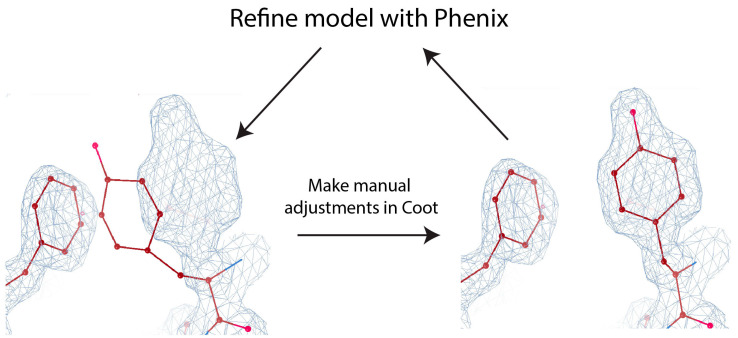
In cases of high-resolution maps, atomic models can be built using tools originally designed for X-ray crystallographers. In an iterative process, the atomic model is refined using Phenix [128], followed by manual adjustments in Coot [130] to address issues such as interatomic clashes or geometry restraints. Here, a tyrosine sidechain rotamer is fit manually into the EM density using Coot.

**Figure 5 micromachines-14-01674-f005:**
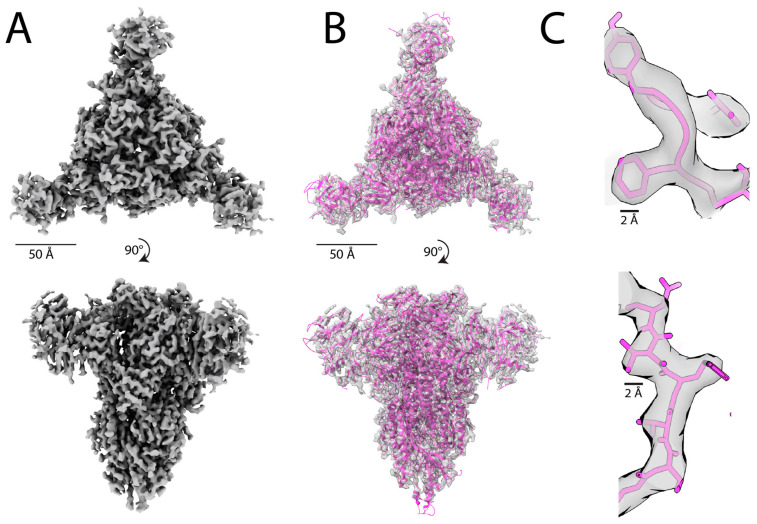
The atomic model built from the cryo-EM map of a feline coronavirus spike protein [162] using DeepTracer [60]. (**A**) The 3.3 Å cryo-EM map of a feline coronavirus spike protein (EMDB ID: EMD-9891) that contains 1403 residues. (**B**) Density map fitted with the DeepTracer model. The DeepTracer model was built in just 14 min, compared to over 60 h required for model building with Phenix [60]. (**C**) Visualization of individual backbone atoms and side chains fitted to the cryo-EM Coulomb map using the molecular model obtained with DeepTracer.

**Table 1 micromachines-14-01674-t001:** Comparison of machine learning and deep learning.

	Machine Learning	Deep Learning
Architecture	Learn data patterns through feature engineering and statistical methods	Learn hierarchal data patterns through NNs
Training Requirements	Requires relatively smaller dataset Less time to train	Requires larger dataset More time to train
Hardware Requirements	Can be trained on standard hardware, CPUs	Requires more powerful hardware, GPUs
Complexity	Less complex, logic easier to understand	More complex, logic harder to understand
Examples	Linear regression, SVM, k-NN, k-means clustering	CNN, U-net, GAN, autoencoder

## Data Availability

No new data were created or analyzed in this study. Data sharing is not applicable to this article.

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
