# Peer review of "Novel Artificial Intelligence-Based Approaches for Ab Initio Structure Determination and Atomic Model Building for Cryo-Electron Microscopy"

_micromachines, 2023, doi:10.3390/mi14091674_

Round 1

Reviewer 1 Report

The authors present a comprehensive overview of the incorporation of AI tools into several pivotal CryoEM challenges, specifically: 1) Ab initio volume reconstruction; 2) de novo atomic building; and 3) protein structure prediction. The manuscript adeptly compiles a majority of the recently introduced methods. As a review, it feels exhaustive and complete. Nonetheless, it might benefit the authors to further their investigation by applying these methods to actual data, thereby establishing a preliminary benchmark of their comparative efficacy. While theoretical presentations in papers often depict methods as highly effective, real-world applications sometimes prove otherwise. Establishing a preliminary benchmark could offer insights into the practical efficacy of these methods. For example, trialing three examples from each category using various programs and sharing the outcomes would be informative. While such a comparison would be highly insightful, the article could still be deemed acceptable even without it, should the authors believe this benchmark isn't suitable for this context.

Author Response

We would like to thank the Reviewer for taking the time to review the manuscript. We appreciate the comments and constructive criticism that gave us the opportunity to revise and improve the manuscript. The concerns raised by the Reviewer are addressed below.

"The authors present a comprehensive overview of the incorporation of AI tools into several pivotal CryoEM challenges, specifically: 1) Ab initio volume reconstruction; 2) de novo atomic building; and 3) protein structure prediction. The manuscript adeptly compiles a majority of the recently introduced methods. As a review, it feels exhaustive and complete. Nonetheless, it might benefit the authors to further their investigation by applying these methods to actual data, thereby establishing a preliminary benchmark of their comparative efficacy. While theoretical presentations in papers often depict methods as highly effective, real-world applications sometimes prove otherwise. Establishing a preliminary benchmark could offer insights into the practical efficacy of these methods. For example, trialing three examples from each category using various programs and sharing the outcomes would be informative. While such a comparison would be highly insightful, the article could still be deemed acceptable even without it, should the authors believe this benchmark isn't suitable for this context."

We thank the Reviewer for kind comments and the suggestion of employing the experimental data sets to establish a preliminary benchmark for comparative efficacy between discussed AI-based algorithms. While we do agree with the Reviewer that such a comparison would be insightful, it extends beyond the scope and formatting constraints of the current review. Thus, we plan to address the Reviewer’s suggestion as the follow-up project in the future.

Reviewer 2 Report

This manuscript summarizes the newly works for ab initio volume generation, heterogeneous 3D reconstruction, and atomic model building from cryo-EM data. I have the following suggestions and comments:

1. It's important to note that model refinement methods based on cryo-electron microscopy data play a crucial role in enhancing the model's quality. The authors should discuss this in the manuscript. For example, the Molecular dynamics-based method MDFF (eLife, 5: e16105, 2016), the conventional energy approach EM-Refiner (Journal of Molecular Biology, 432(19): 5365-5377, 2023), and Flex-EM (Structure, 16: 673-683, 2008) can be added in Section 5, and the title can be modified to “Conventional approaches to de novo atomic model building & refinement”.

 2. Some relevant work has been overlooked. For example, DEMO-EM (Nature Comput. Sci., 2(3): 265-275) presented by Zhang Group is also a model building method from cryo-EM density maps, which uses AI predicted distance to guide the modeling. I strongly suggest add it to Table 3 or Section 7 and briefly introduce it.

 3. All methods enumerated in Table 2 seem to exhibit analogous functionality. To enhance clarity, I suggest that the authors present their respective advantages just like that in Table 3.

 4. AI-based protein structure prediction is crucial for cryo-electron microscopy modeling. Several other works should be introduced in this Section. For example, the I-TASSER-MTD (Nature Protocols, 17: 2326-2353, 2022) also supports cryo-EM data-assisted modeling. In addition, it would be better if the authors could add some results in Section 7.

NO

Author Response

We would like to thank the Reviewer for taking the time to review the manuscript. We appreciate the comments and constructive criticism that gave us the opportunity to revise and improve the manuscript. The concerns raised by the Reviewer are addressed below.

  1. "It's important to note that model refinement methods based on cryo-electron microscopy data play a crucial role in enhancing the model's quality. The authors should discuss this in the manuscript. For example, the Molecular dynamics-based method MDFF (eLife, 5: e16105, 2016), the conventional energy approach EM-Refiner (Journal of Molecular Biology, 432(19): 5365-5377, 2023), and Flex-EM (Structure, 16: 673-683, 2008) can be added in Section 5, and the title can be modified to “Conventional approaches to de novo atomic model building & refinement”."

We thank the Reviewer for pointing out this important omission. The model refinement methods and software, including MDFF, EM-Refiner, and Flex-EM are now discussed on page 14 in the third paragraph of section 5 (lines 493-509), and properly referenced in the manuscript. All changes are highlighted in red in the revised manuscript. In addition, the title of section 5 has been modified as advised by the Reviewer.

The following new references have been added to the revised manuscript:

  1. Trabuco, L.G.; Villa, E.; Schreiner, E.; Harrison, C.B.; Schulten, K. Molecular dynamics flexible fitting: a practical guide to combine cryo-electron microscopy and X-ray crystallography. Methods 2009, 49, 174-180. 
  2. Singharoy, A.; Teo, I.; McGreevy, R.; Stone, J.E.; Zhao, J.; Schulten, K. Molecular dynamics-based refinement and validation for sub-5 Å cryo-electron microscopy maps. Elife, 2016, 5, e16105. 
  3. Dang, S.; Feng, S.; Tien, J.; Peters, C.J.; Bulkley, D.; Lolicato, M.; Zhao, J.; Zuberbühler, K.; Ye, W.; Qi, L.; Chen, T. Cryo-EM structures of the TMEM16A calcium-activated chloride channel. Nature 2017, 552, 426-429.
  4. Yao, H.; Song, Y.; Chen, Y.; Wu, N.; Xu, J.; Sun, C.; Zhang, J.; Weng, T.; Zhang, Z.; Wu, Z.; Cheng, L. Molecular architecture of the SARS-CoV-2 virus. Cell 2020, 183, 730-738.
  5. Mansbach, R.A.; Chakraborty, S.; Nguyen, K.; Montefiori, D.C.; Korber, B.; Gnanakaran, G. The SARS-CoV-2 Spike variant D614G favors an open conformational state. Biophys. J .2021, 120, 298a.
  6. Topf, M.; Lasker, K.; Webb, B.; Wolfson, H.; Chiu, W.; Sali, A. Protein structure fitting and refinement guided by cryo-EM density. Structure 2008, 16, 295-307.
  7. Zhang, B.; Zhang, X.; Pearce, R.; Shen, H.B.; Zhang, Y. A new protocol for atomic-level protein structure modeling and refinement using low-to-medium resolution Cryo-EM density maps. J. Mol. Biol. 2020, 432, 5365-5377. 

  1. "Some relevant work has been overlooked. For example, DEMO-EM (Nature Comput. Sci., 2(3): 265-275) presented by Zhang Group is also a model building method from cryo-EM density maps, which uses AI predicted distance to guide the modeling. I strongly suggest add it to Table 3 or Section 7 and briefly introduce it."

We apologize for this important omission. DEMO-EM is now discussed in the third paragraph of section 7 on page 20 (lines 696-701). Furthermore, DEMO-EM is listed in the Table 3 on page 19 in the revised manuscript.

  1. "All methods enumerated in Table 2 seem to exhibit analogous functionality. To enhance clarity, I suggest that the authors present their respective advantages just like that in Table 3."

We reformatted the Table 2. The table now lists respective benefits and limitations for each and every discussed software package.

  1. "AI-based protein structure prediction is crucial for cryo-electron microscopy modeling. Several other works should be introduced in this Section. For example, the I-TASSER-MTD (Nature Protocols, 17: 2326-2353, 2022) also supports cryo-EM data-assisted modeling. In addition, it would be better if the authors could add some results in Section 7."

We thank the Reviewer for this comment. We now discuss I-TASSER-MTD in the second paragraph of the section 7 on page 19 (lines 669-674). In addition, as stated above in the response to one of the Reviewer’s previous comments, we introduce a related program DEMO-EM (page 20, lines 696-701), along with EMBUILD (page 20, lines 687-696),

Following the Reviewer’s suggestion, the section 7 now includes statements highlighting some of the results obtained with different modeling programs. For example:

-page 20, lines 693-696

“He et al. applied EMBUILD to 47 intermediate-resolution maps and built models with an average template modeling score (TM-score) of 0.909 and RMSD of 2.85 Å, outperforming other state-of-the-art methods, including phenix.dock_in_map and gmfit.”

-page 20, lines 698-701

“When applied to a benchmark set of multi-domain proteins with medium- to low-resolution maps, DEMO-EM generated models with correct inter-domain orientations in 97% of cases, outperforming MDFF and Rosetta.”

-page 20, lines 712-713

“In a blind test of 13 experimental maps, DeepTracer-ID identified the correct proteins as top candidates”

We also added the following references to the revised manuscript:

  1. Zhou, X.; Li, Y.; Zhang, C.; Zheng, W.; Zhang, G.; Zhang, Y., Progressive assembly of multi-domain protein structures from cryo-EM density maps. Nat. Comp. Sci. 2022, 2, 265-275.
  2. Zhou, X.; Zheng, W.; Li, Y.; Pearce, R.; Zhang, C.; Bell, E.W.; Zhang, G.; Zhang, Y. I-TASSER-MTD: a deep-learning-based platform for multi-domain protein structure and function prediction. Nat. Protoc. 2022, 17, 2326-2353.

Reviewer 3 Report

The authors provide a detailed review that compares traditional methods to DL ones for cryoEM reconstruction and atomic model building. This will be beneficial for new members joining the cryoEM community. There are some missing points that if addressed can help clarify the review:

  • Include subtitles on Fig. 3.
  • Include a discussion about the methods to analyze conformational heterogeneity that do to not use VAEs but rather use some type of dimensionality reduction, for example, Zernike polynomials (Herreros et al. Nat Commun 2023). 
  • Include a discussion about the methods that go from images to atomic models directly and or vice versa to study conformational probability distributions (e.g., Tang JPCB 2023). 
  • Include a discussion about DL methods that combine molecular dynamics information with cryo-EM map variability (e.g., Matsumoto et al. Nature Machine Intelligence 2021)
  • A general discussion about combing AI with cryoEM is missing. What metrics for validating conformational landscapes and atomic models from AI should people use? What should people be cautious about? 
  • Additional refs that might be missing:
    • Nashed et al. https://arxiv.org/abs/2107.02958
    • General review: Tang et al. COSB 2023. 

Author Response

We would like to thank the Reviewer for taking the time to review the manuscript. We appreciate the comments and constructive criticism that gave us the opportunity to revise and improve the manuscript. The concerns raised by the Reviewer are addressed below.

  1. “Include subtitles on Fig. 3.”

The legend of Figure 3 now contains subtitles.

  1. “Include a discussion about the methods to analyze conformational heterogeneity that do to not use VAEs but rather use some type of dimensionality reduction, for example, Zernike polynomials (Herreros et al. Nat Commun 2023).“

We thank the Reviewer for this suggestion. We have expanded the third paragraph of section 4 by including several statements describing methods that utilize dimensionality reduction, such as Zernike polynomials. Please see highlighted text on page 10, lines 346-360.

In addition, the following references are cited in the revised manuscript:

  1. Liu, W.; Frank, J. Estimation of variance distribution in three-dimensional reconstruction. I. Theory. J. Opt. Soc. Am. A 1995, 12, 2615–2627.
  2. Penczek, P.A.; Kimmel, M.; Spahn, C.M. Identifying conformational states of macromolecules by eigen-analysis of resampled cryo-EM images. Structure 2011, 19, 1582–1590.
  3. Tagare, H.D.; Kucukelbir, A.; Sigworth, F.J.; Wang, H.; Rao, M. Directly reconstructing principal components of heterogeneous particles from cryo-EM images. J. Struct. Biol. 2015, 191, 245–262.
  4. Nakane, T.; Kimanius, D.; Lindahl, E.; Scheres, S.H. Characterisation of molecular motions in cryo-EM single-particle data by multi-body refinement in RELION. Elife 2018, 7, e36861.
  5. Jin, Q.; Sorzano, C.O.S.; De La Rosa-Trevín, J.M.; Bilbao-Castro, J.R.; Núñez-Ramírez, R.; Llorca, O.; Tama, F.; Jonic, S. Iterative elastic 3d-to-2d alignment method using normal modes for studying structural dynamics of large macromolecular complexes. Structure 2014, 22, 496–506. 
  6. Hamitouche, I.; Jonic, S. DeepHEMNMA: ResNet-based hybrid analysis of continuous conformational heterogeneity in cryo-EM single particle images. Front. Mol. Biosci. 2022, 9, 965645.
  7. Herreros, D.; Lederman, R.R.; Krieger, J.M.; Jiménez-Moreno, A.; Martínez, M.; Myška, D.; Strelak, D.; Filipovic, J.; Sorzano, C.O.S.; Carazo, J.M. Estimating conformational landscapes from Cryo-EM particles by 3D Zernike polynomials. Nat. Commun. 2023, 14, 154.
  8. Herreros, D.; Lederman, R.R.; Krieger, J.M.; Jiménez-Moreno, A.; Martínez, M.; Myška, D.; Strelak, D.; Filipovic, J.; Bahar, I.; Carazo, J.M; Sanchez, C.O.S. Approximating deformation fields for the analysis of continuous heterogeneity of biological macromolecules by 3D Zernike polynomials. IUCrJ 2021, 8, 992-1005.
  9. Lederman, R.R.; Andén, J.; Singer, A. Hyper-molecules: on the representation and recovery of dynamical structures for applications in flexible macro-molecules in cryo-EM. Inverse Probl. 2020, 36, 044005.
  10. Singer, A.; Sigworth, F.J. Computational methods for single-particle electron cryomicroscopy. Ann. Rev. Biomed. Data Sci. 2020, 3,163-190.

  1. “Include a discussion about the methods that go from images to atomic models directly and or vice versa to study conformational probability distributions (e.g., Tang JPCB 2023).”

The discussion related to inferring atomic models from the experimental images and subsequent characterization of the conformational probability distributions has been added to the seventh paragraph of section 4 on page 11, (lines 411-429).

Also, additional references have been included in the revised manuscript:

  1. Cossio, P.; Hummer, G. Bayesian analysis of individual electron microscopy images: Towards structures of dynamic and heterogeneous biomolecular assemblies. J. Struct. Biol. 2013, 184, 427-437.
  2. Cossio, P.; Rohr, D.; Baruffa, F.; Rampp, M.; Lindenstruth, V.; Hummer, G. BioEM: GPU-accelerated computing of Bayesian inference of electron microscopy images. Comp. Phys. Commun. 2017, 210, 163-171.
  3. Giraldo-Barreto, J.; Ortiz, S.; Thiede, E.H.; Palacio-Rodriguez, K.; Carpenter, B.; Barnett, A.H.; Cossio, P. A Bayesian approach to extracting free-energy profiles from cryo-electron microscopy experiments. Sci. Rep. 2021, 11, 13657.
  4. Tang, W.S.; Silva-Sánchez, D.; Giraldo-Barreto, J.; Carpenter, B.; Hanson, S.; Barnett, A.H.; Thiede, E.H.; Cossio, P. Ensemble reweighting using Cryo-EM particles. J. Phys. Chem. B 2023, 127, 4510-5421.

  1. “Include a discussion about DL methods that combine molecular dynamics information with cryo-EM map variability (e.g., Matsumoto et al. Nature Machine Intelligence 2021).”

We thank the Reviewer for this suggestion. We have added a discussion concerned with methods that combine MD simulations with cryo-EM data to the sixth paragraph of section 4 on pages 10-11, (lines 391-410).

We also included the following references in the revised manuscript:

  1. Van Den Bedem, H.; Fraser, J.S. Integrative, dynamic structural biology at atomic resolution—it's about time. Nat. Methods 2015, 12, 307-318.
  2. Bonomi, M.; Pellarin, R.; Vendruscolo, M. Simultaneous determination of protein structure and dynamics using cryo-electron microscopy. Biophys. J. 2018, 114, 1604-1613.
  3. Tang, W.S.; Zhong, E.D.; Hanson, S.M.; Thiede, E.H; Cossio, P. Conformational heterogeneity and probability distributions from single-particle cryo-electron microscopy. Cur. Opin. Struct. Biol. 2023, 81, 102626.
  4. Matsumoto, S.; Ishida, S.; Araki, M.; Kato, T.; Terayama, K.; Okuno, Y. Extraction of protein dynamics information from cryo-EM maps using deep learning. Nat. Mach. Intell. 2021, 3, 153-160.

  1. “A general discussion about combing AI with cryoEM is missing. What metrics for validating conformational landscapes and atomic models from AI should people use? What should people be cautious about?”

An expanded general discussion concerned with combining AI with cryo EM is now presented in the section 8 on page 21 (lines 747-762). In brief, we highlight the lack of stand-alone validation metrics for newly proposed techniques and advise validation of the results against experimentally-derived data. To avoid redundancy, analogous statements were removed from sections 4 and 7.

  1. “Additional refs that might be missing:
    • Nashed et al. https://arxiv.org/abs/2107.02958”

We thank the Reviewer for this suggestion, however the provided link refers to a pre-print version of the article, which is already cited throughout the manuscript. For example, in section 4 on page 9 (line 303), and in Table 2 (page 12, line 451).

The following reference is included in the manuscript:

  1. Nashed, Y.; Poitevin, F.; Guan, H.; Woollard, G.; Kagan, M.; Yoon, C.H.; Ratner, D. CryoPoseNet: end-to-end simultaneous learning of single-particle orientation and 3D map reconstruction from cryo-electron microscopy data. In Proceedings of the International Conference on Computer Vision (ICCV), Montreal, QC, Canada, 11-17 October 2021; pp. 4066-4076.

  • “General review: Tang et al. COSB 2023.”

We apologize for omitting this important reference in the manuscript, however the abovementioned article was published while our manuscript was under review. The article by Tang et al. is now referred to in the sixth and seventh paragraphs of the section 4 on page 11 (lines 410, 426 and429) and cited as:

  1. Tang, W.S.; Zhong, E.D.; Hanson, S.M.; Thiede, E.H; Cossio, P. Conformational heterogeneity and probability distributions from single-particle cryo-electron microscopy. Cur. Opin. Struct. Biol. 2023, 81, 102626.